# Numerical Analysis for Light Absorption Spectra of the Base of DNA-Wrapped Single-Walled Carbon Nanotubes

**DOI:** 10.3390/molecules28062719

**Published:** 2023-03-17

**Authors:** Hisao Taira, Daisuke Miyashiro, Kazuo Umemura

**Affiliations:** 1Faculty of Education, Hokkaido University of Education, 5-3-1-5, Ainosato, Kita-ku, Sapporo 002-8502, Japan; 2Department of Physics, Graduate School of Science, Tokyo University of Science, 1-3, Kagurazaka, Shinjuku-ku, Tokyo 162-8601, Japan

**Keywords:** carbon nanotube, DNA, light absorption spectra, numerical calculation

## Abstract

This study numerically demonstrates the light absorption spectra of each base of DNA-wrapped single-walled carbon nanotubes (SWCNTs). Previous experimental and theoretical studies show that the optical properties of these composites are different from the bare SWCNTs. In this work, we investigated the bases of DNA that influence optical properties. To obtain stable molecular states for studying optical properties, molecular dynamics calculations were performed. Additionally, light absorption spectra in the ultraviolet-to-near-infrared region of one type of base-wrapped (e.g., adenine-, thymine-, cytosine-, or guanine-wrapped) SWCNTs were investigated by utilizing the semi-empirical molecular orbital theory using SCIGRESS commercial software. This method can significantly reduce the calculation time compared to the ab initio molecular orbital method, making the handling of composites of bases and SWCNTs possible. We found that the largest peaks appear at a wavelength of around 300 nm for all the composites. Furthermore, we found that the light absorption spectra above 570 nm are strongly influenced by adenine and cytosine. Thus, our computational results provide insight into the optical properties and the effects of base–SWCNTs that are difficult to investigate experimentally under the influence of solvents and various molecules.

## 1. Introduction

Single-walled carbon nanotubes (SWCNTs) have potential applications in various industries, such as electronics, nano mechanics, health care, etc. SWCNTs are also useful nanomaterials for biotechnology, aerogels, films, fibers, and energy supply systems [1,2,3]. SWCNTs are dispersed in plant-derived nanocellulose and DNA, which have little impact on the human body [4]. In particular, there are several reports indicating the successful production of single-stranded DNA-wrapped SWCNTs (ssDNA–SWCNTs) [5,6,7]. Stable dispersion of ssDNA–SWCNTs in water offers potential uses as biomaterials. The stable interaction between DNA strands and SWCNTs implies that the DNA–SWCNTs binding energy is large and results in stable nanotubes in the external environment.

On the other hand, from a macro perspective, our previous research focused on the engineering application of DNA–SWCNT as a biomaterial and analyzed the mechanical vibration properties using the finite element method (FEM) [8,9]. The incorporation of SWCNTs into cells according to their length was also investigated, considering their use as drug delivery and biosensors. In our previous work, we used FEM to predict the mechanical behavior of DNA–SWCNTs with different lengths attached to cells in water [10]. To safely and effectively demonstrate the functions of DNA–SWCNTs, it is necessary to understand the physical, chemical, and biological properties of DNA–SWCNTs. Dispersion of SWCNTs is possible not only in single-stranded DNA but also in double-stranded DNA (dsDNA), and research is being conducted to examine the optical absorption spectral characteristics to investigate the properties of DNA–SWCNTs as biosensors [11]. The difference in the light absorption spectra of ssDNA–SWCNT and dsDNA–SWCNT is believed to be due to the difference in the structure of wrapped ssDNA and dsDNA. There is also a consideration that ssDNA has more exposed bases than dsDNA, making it more stable and easier to adsorb to the SWCNT surface. On the other hand, dsDNA–SWCNTs are cheaper, can be easily dispersed, and are often used in experimental studies. There are experimental studies that have shown the light absorption spectra of the dsDNA-(6,5) nanotube depending on which solution it is in. These previous studies compared the light absorption spectra of two materials: SWCNTs and DNA wrapped around them. They show that in the case of the composites, there is a shift in the position of the peak and the appearance of a new broad peak. This means that the light absorption spectra change due to the wrapping of the DNA. Since DNA comprises four bases, adenine, thymine, cytosine, and guanine, understanding their dominance in the light absorption spectra is critical. This means that the optical absorption spectra of one type of base-wrapped SWCNTs, namely, adenine-, thymine-, cytosine-, or guanine-wrapped SWCNTs, are essential to investigate. However, it is difficult to experimentally obtain optical absorption spectra between SWCNTs and the base. The powder preparation method, solution conditions (reagent composition, pH, ionic strength), and measurement errors affect the optical absorption spectra of SWCNTs (reagent composition, pH, ionic strength). At this point, there are some theoretical and numerical studies on the optical absorption spectra of composites of SWCNTs [12,13,14,15]. However, few studies investigated the optical absorption spectra between SWCNTs and bases. Although it has been shown that both single-stranded and double-stranded DNA can be adsorbed and dispersed on SWCNTs, it is difficult to investigate these mechanisms only by experimental approaches due to the effects of composite interactions between solvent and DNA. Therefore, simple states between each base and SWCNTs were created and investigated numerically.

Using molecular dynamics methods, this study used SCIGRESS in commercial software (Version 3.5, Fujitsu Ltd, Tokyo, Japan.) to calculate the stable state of SWCNTs wrapped with adenine, thymine, cytosine, or guanine. Further, we calculated the light absorption spectra of these composites over a wide range from ultraviolet to near-infrared light by the semi-empirical molecular orbital theory method. The light absorption spectra calculated by the theory provide the optical properties of each base and SWCNTs under vacuum, omitting the complicated conditions of the experimental solution. We also investigated the effect of each number of bases placed around SWCNTs on their optical absorption spectra. These results are expected to contribute to a fundamental understanding of the optical properties of DNA–SWCNTs composites.

## 2. Results

Molecular arrangement for six base numbers of all targeted adenine-, thymine-, cytosine-, or guanine-wrapped SWCNTs composites are prepared by SCIGRESS. It is important to note that the molecular arrangement for each base number is determined by the molecular dynamics method. Figure 1 shows the molecular model of adenine-wrapped SWCNTs as a representative model. The distances between SWCNTs and bases are about 0.4295 nm, 0.4542 nm, 0.4636 nm, and 0.4411 nm for adenine, thymine, cytosine, and guanine, respectively. Distances between bases are 0.8828 nm, 0.6968, 0.6643 nm, and 0.6518 nm in the case of adenine, thymine, cytosine, and guanine, respectively. The length of the SWCNTs was fixed at 1 nm, and the bases around it were set so that they would not touch each other, with a maximum of six bases.

First, we compared the light absorption spectra of three types of bare SWCNTs and base and SWCNTs wrapped by the base. Figure 2, Figure 3, Figure 4 and Figure 5 show the wavelength dependences of light of the molar absorbance coefficients of bases (black), bare (6,5) nanotubes (blue), and (6,5) nanotubes wrapped by one base (green). In these figures, for example, 6_5_0 indicates the (6,5) CNT with no attached base, 6_5_1 the (6,5) CNT with one attached base, and others. Common features of all figures are that the most significant peaks for bare SWCNTs appear at a wavelength of around 300 nm, comparable to the observed peak at 260 nm in the experiments [16]. In addition, the high amplitude level maintained in the range of 750–1000 nm is also consistent with our experimental results.

The wavelength peaks of SWCNTs calculated differ by about 15% compared to the experimental results. We believe that the differences are due to experimental and computational conditions variations. All light absorption spectra in our studies were performed in a vacuum, but the experiments were performed in water. This difference implies that the dielectric background of water affects the exciton optical transitions but not the electric dipole–vibronic transitions. Therefore, the effect of the solvent is significant. Moreover, previous studies report that the pH and ionic strength of water also change the light absorption spectra [17], which may lead to differences between experiments and calculations. In addition, it cannot be denied that the SWCNTs used in the experiments were contaminated with impurities during manufacturing, leading to measurement errors. On the other hand, it can be confirmed that the light absorption spectrum of each base is mainly in the ultraviolet region below 300 nm, as indicated by the black lines in Figure 2, Figure 3, Figure 4 and Figure 5. Since the light absorption spectrum of DNA bases generally exists around 260 nm, it can be verified that our calculation results are generally valid. The most interesting aspect of this calculation result is that adenine and cytosine affect the near-infrared region above 800 nm. Since it is difficult to create an experimental environment for verifying such behavior, this is a significant result obtained in the calculations of this work.

Figure 6, Figure 7, Figure 8 and Figure 9 show the variation in the molar absorbance coefficient as a function of the wavelength of light for adenine-, thymine-, cytosine-, or guanine-wrapped-SWCNTs. Let us reveal the characteristics of each figure. Figure 6 shows the wavelength dependence of light of the molar absorbance coefficient of adenine-wrapped SWCNTs. The number of attached adenines is from one to six. This graph has two features. Firstly, the value of the peak at around 300 nm increases or decreases depending on the number of adenines. Specifically, it becomes larger when the number of adenines is one to four and smaller when the number of adenines is five to six. Another attribute is that the profiles are similar when the number of bases is one or three and similar in other cases. Furthermore, two new peaks appear in the near-infrared region above 800 nm.

Figure 7 shows the wavelength dependence of light of the molar absorbance coefficient of thymine-wrapped SWCNTs. In this case, there is no change in the optical absorption spectra even if the number of bases increases. This means that the interaction between thymine and SWCNT is minimal compared to SWCNTs or thymine alone.

Figure 8 shows the wavelength dependence of light of the molar absorbance coefficient of cytosine-wrapped SWCNTs. Two characteristics are important to note in this graph. The first is that the peak value at around 300 nm becomes larger or smaller depending on the number of bases bound. The other characteristic is that the graph profile is similar for all the bases between one and five. However, the profile is different for zero and six bases. Furthermore, one new peak appears in the near-infrared region above 700 nm.

Figure 9 shows the wavelength dependence of light of the molar absorbance coefficient of guanine-wrapped SWCNTs. The three important aspects in this graph are as follows: first is that the peak at around 300 nm splits into two peaks; second, the peak at around 570 nm moves to a larger wavelength; and thirdly, the magnitude of these moved peaks becomes larger.

Figure 10 shows the wavelength dependence of light of the molar absorbance coefficient of bare-, one adenine-, one thymine-, one cytosine-, or one guanine- wrapped SWCNTs. Cytosine strongly affects the wavelength dependence of light of the molar absorbance coefficient. In addition to the most significant peaks around 300 nm, further 575 nm and 875 nm peaks appear.

## 3. Discussion

The present study numerically calculated the wavelength dependence of light of the molar absorbance coefficient of the adenine-, thymine-, cytosine-, or guanine-wrapped SWCNTs. The most crucial finding is that the largest peaks appear at a wavelength of around 300 nm for the case of all composites. This also proves that the numerical simulation is in good agreement with the experimental data, and the peak value of the wavelength does not shift due to the interaction between bases and SWCNTs. The molar absorbance coefficients of adenine and cytosine, indicated by the black lines in Figure 2, Figure 3, Figure 4 and Figure 5, are smaller than those of thymine and guanine. On the one hand, it is interesting that the effects of adenine and cytosine on the molar absorbance coefficients when wrapped in SWCNTs are greater than those of thymine and guanine. As shown in Figure 10, the adenine- and cytosine-wrapped SWCNTs exhibit significant spectral changes beyond 570 nm compared to bare SWCNTs. This result can be expected to support the explanation of the cause of the absorption spectra change of DNA–SWCNTs shown in the previous research [11]. The experimental results of the light absorption spectra obscure the intrinsic properties of the structure of DNA–SWCNTs due to the composite effects of solvents. In this study, we can present the absorption spectra in the state where only one base is arranged around the SWCNTs, omitting the effect of water molecules. We propose a basic analytical result for interpreting the experimental results. Numerical calculation results, such as the semi-empirical molecular orbital method, differ from actual experimental results because molecular interactions of solvents are not considered. Although it may not be possible to quantitatively compare the calculated results of base–SWCNTs composites, which is the subject of this study, with the experimental results, the calculated light absorption spectra are effectively used to discuss the mechanism of the phenomena occurring in the experimental results. In fact, preparing composites with only specific bases wrapped around SWCNTs in experiments is quite challenging. The advantage of our research is that we can obtain such intermolecular effects based on theoretical calculations.

An attempt has been made to understand why thymine wrapping has a much weaker effect on the light absorption spectra than other base wrapping. The difference in structure with other bases is affected as the spectra do not change when the thymine is rearranged with respect to SWCNTs. For example, as cytosine and thymine are considered, they have the same six-membered ring structure. However, the difference is that thymine has more oxygen attached, so thymine has fewer electrons taken away, resulting in no change in the spectra. Compared to guanine, which has the next smallest spectral change after thymine, guanine has the same six-membered ring structure as thymine in the way that the addition of a five-membered ring takes up electrons. Although the results presented in this study cannot give a clear interpretation of the effect of the structure of each base on SWCNTs, we could calculate the effect of the interaction of each base with SWCNTs on the spectra. We believe that this contributes to the understanding of future experimental results. These findings are expected to accelerate the study of optical characteristics of base-wrapped DNA–SWCNTs. Furthermore, the previous experimental study showed that the peak of the molar absorbance coefficient was observed near 990 nm [11]. In the previous study, redox reactions were known to change the light absorption spectra, but the cause had not been identified experimentally. Although these conditions were difficult to reproduce in our computational model, we were able to investigate how individual bases affect the light absorption spectra of SWCNTs. For example, the light absorption spectra of SWCNTs calculated are sensitive to the adsorption of adenine and cytosine in the region above 570 nm. These results are still affected by increasing the number of adenines and cytosines, as shown in Figure 6 and Figure 8. These results suggest the possibility that adenine and cytosine influence the spectra change when a redox agent is added to the DNA–SWCNTs composites. We recognize that these results do not allow us to conclude the mechanism of the experimental results. In order to clarify these factors, we believe it is effective to complement both the experimental research and the calculation results that do not consider the solvent shown in this research. This previous study was performed during redox, and there is the solvent. As a potential application, a candidate of green molecules in the field of energy supply systems, redox and solvent are important factors. It is expected that research will continue to advance through both experiments and numerical analysis approaches.

Finally, we present future work to further develop this study. The semi-empirical molecular orbital theory method used in this study is a practical method that significantly reduces the computation time compared to the first principles molecular orbital method. However, obtaining the optical absorption spectra of bases and SWCNTs up to near-infrared light requires much calculation time. To solve these problems, material informatics using deep learning may be useful. In research on the light absorption spectra of DNA–SWCNT, a method of creating a Bayesian regularized back-propagation neural network model from a small number of experimental results and predicting various conditions in a data-driven manner has been reported [18]. Based on the molecular orbital calculation results obtained in this research, the various conditions are expected to be predicted efficiently using a Bayesian regularization back-propagation neural network model based on representative calculation results. These methods are needed because our molecular models are still missing essential factors. For example, a remaining issue in this study is the effect of SWCNT chirality and length.

In this work, we fix it at 1 nm and show only calculations limited to (6,5) SWCNTs. Countless combinations of chirality and lengths are associated with SWCNTs, and it is difficult to calculate all combinations by the molecular orbital method. In addition, only bases were wrapped in SWCNTs. However, in the future, it will be necessary to investigate interactions with other molecules, such as deoxyribonucleic acid and phosphoric acid, and the effects of the state in which multiple bases are linked. Furthermore, there is no doubt that the influence of water molecules around those DNA–SWCNTs is also a crucial factor to investigate. Reproducing these effects with a molecular model will result in a huge molecular weight, and the computation time is predicted to increase significantly. It is likely that the number of cases that can actually be calculated is limited. In this context, machine learning may be required to predict various conditions from a small amount of data. Therefore, future work is expected to utilize machine learning, such as the Bayesian regularized back-propagation neural network model.

## 4. Methods

The light absorption spectra of bases of DNA and SWCNTs presented in this study were performed using the molecular dynamics and molecular orbital algorithms provided in SCIGRESS basic (Fujitsu Co. Ltd.). This software is developed based on widely used molecular orbital theory [19,20] and numerical calculation theory [21], and is used in various physical and chemical research. Every calculation was performed under the vacuum condition. The SWCNTs model composed of six-membered rings was set to (6,5) chirality using the CNT-builder provided by SCIGRESS. The bases, adenine, guanine, cytosine, and thymine, used molecular models provided in the SCIGRESS library. The molecular model of SWCNTs was set to (6,5) chirality using CNT-builder. The length was set to 1 nm, which is the length that allows multiple DNA bases to be arranged without overlapping. The length of SWCNTs was set to the minimum necessary to prevent an increase in computation time. The initial positions in the molecular model of the SWCNTs and bases created by SCIGRESS do not consider intermolecular interactions. Since the light absorption spectra results are greatly affected by intermolecular interactions, it is necessary to create a molecular configuration that considers these effects between SWCNTs and bases. The stable state between the bases and SWCNTs when calculating the light absorption spectra was determined by solving an optimization problem that minimizes the Lennard–Jones potential U, as shown as
U = D_0_{(R_0_/r)^12^ − 2(R_0_/r)^6^}(1)
where D_0_ and R_0_ are fitting parameters given by D_0_ =0.2483 kcal/mol and R_0_ = 3.848 Å, respectively. This potential function is widely used since it can express intermolecular interactions simply and accurately. The states with and without consideration of the Lennard–Jones potential are shown in Figure 11a,b, respectively. The molecular arrangement is determined by minimizing this Lennard–Jones potential U. After obtaining the optimized composites, we calculated the light absorption spectra using SCIGRESS basic.

The light absorption spectra were calculated using the molecular orbital method for the molecular model optimized by the Lennard–Jones potential. According to standard molecular orbital theory, an orbital of electrons $\varphi_{i}$ was approximated as a linear combination of atomic orbitals $\chi_{\mu }$,
(2)$$\varphi_{i}=\overset{n}{\underset{\mu =1}{\sum }}C_{\mu i}\chi_{\mu }$$
where $C_{\mu i}$ is a molecular orbital coefficient, and *n* is the total number of atomic orbitals.

There are three types of molecular orbital methods: ab initio molecular orbital method, semi-empirical molecular orbital, and empirical molecular orbital. In this work, we used the semi-empirical molecular orbital theory, the Hartree–Fock approximation, to find $\chi_{\mu }$,. This method used the Hartree–Fock equation with empirical parameters to approximate the electronic states of molecules. Compared to the ab initio molecular orbital method, the amount of calculation is significantly reduced, so it is advantageous for handling large molecules such as SWCNTs and DNA bases treated in this research. In addition, some of the electron correlation effects can be included by using empirical parameters. The semi-empirical method uses the Slater orbital (STO) [22]. This method differs from the ab initio molecular orbital method approach in that only the bases representing the valence orbitals are used in a semi-empirical manner, e.g., for hydrogen, only 1 s were used and for carbon, only 2 s, 2 px, 2 py, and 2 pz types were used in the calculation.

In the semi-empirical method, the zero differential overlap (ZDO) approximation is the usual approach, and the overlap integral matrix becomes just the identity matrix. The Hartree–Fock–Roothaan (HFR) equation is simplified as shown below.
(3)FC=CE

This version of the equation forms the starting point for the semi-empirical method. Solving this equation gives the molecular orbital coefficient *C* and the orbital energy *E*. As a computational procedure, first, the initial value of *C* was computed using a simpler method such as Huckel’s method. This initial guess gave *F*. Then *F* was diagonalized to compute *C* and *E*. This new *C* matrix was used to compute *F* for the next computation of *C* and *E*. Finally, an iterative calculation called self-consistent field (SCF) was performed until *C* converged to satisfy the threshold. By solving this HFR equation, the total electron energy *E* could be obtained. *E* was given by the following formula:(4)$$E=\frac{1}{2}\overset{nbasis}{\underset{\mu ,\upsilon }{\sum }}P_{\mu \upsilon }\left(H_{\mu \upsilon }+F_{\mu \upsilon }\right)$$
where $H_{\mu \upsilon }$ is the matrix element representing the energy corresponding to electron motion, nuclear attraction, and repulsion. *nbasis* is the number of basis functions. $P_{\mu \upsilon }$ is the density matrix. $F_{\mu \upsilon }$ is the Fock matrix, the electron repulsion integral of which is the most computationally expensive part. This calculation used the Zerner’s intermediate neglect of differential overlap (ZINDO) method to evaluate the electron repulsion integral to obtain the electron excited state. From the above, the light absorption spectra in the UV–visible region between SWCNTs and bases shown in this study were calculated.

## Figures and Tables

**Figure 1 molecules-28-02719-f001:**
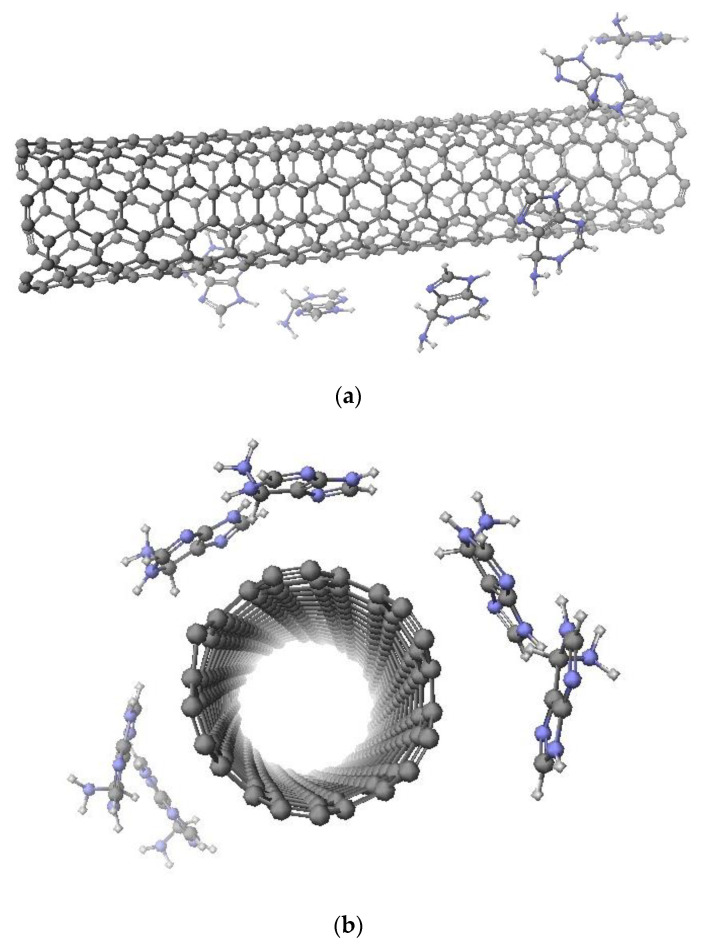
Molecular arrangement of (6,5) nanotube wrapped by six adenines. (**a**) Side view; (**b**) top view.

**Figure 2 molecules-28-02719-f002:**
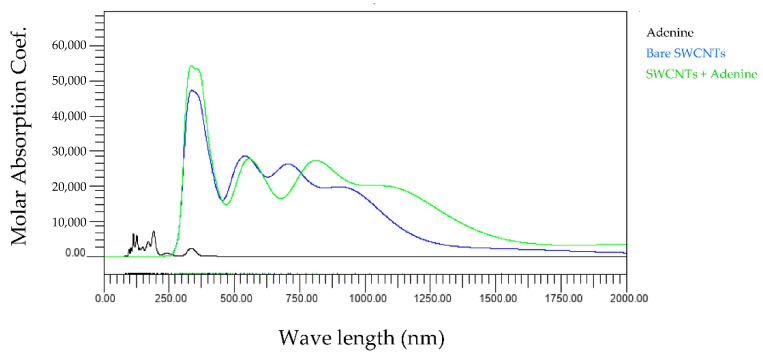
Wavelength dependence of light of the molar absorbance coefficient of adenine (black), bare (6,5) nanotube (blue), and (6,5) nanotube wrapped by one adenine (green).

**Figure 3 molecules-28-02719-f003:**
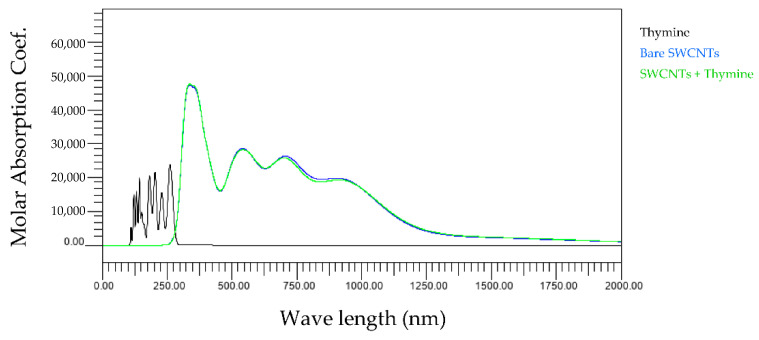
Wavelength dependence of light of the molar absorbance coefficient of thymine (black), bare (6,5) nanotube (blue), and (6,5) nanotube wrapped by one thymine (green).

**Figure 4 molecules-28-02719-f004:**
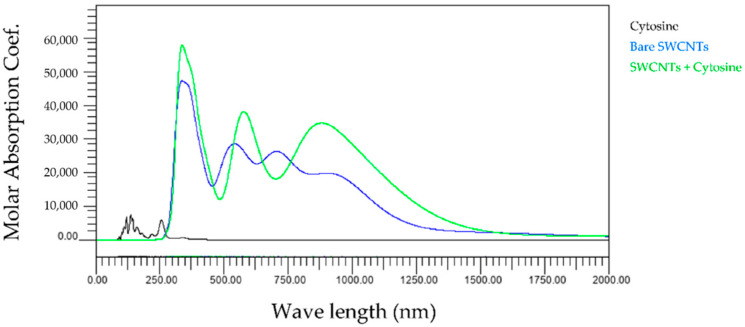
Wavelength dependence of light of the molar absorbance coefficient of cytosine (black), bare (6,5) nanotube (blue), and (6,5) nanotube wrapped by one cytosine (green).

**Figure 5 molecules-28-02719-f005:**
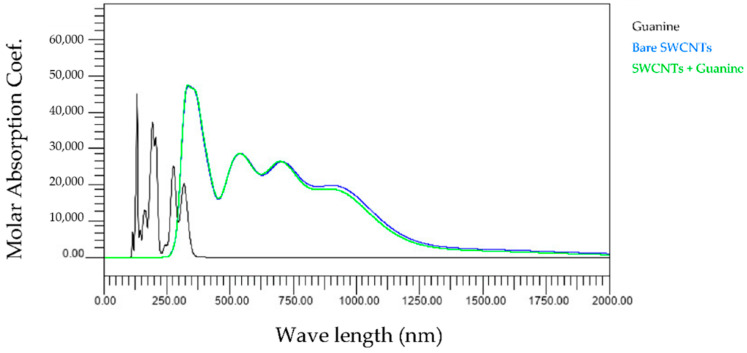
Wavelength dependence of light of the molar absorbance coefficient of guanine (black), bare (6,5) nanotube (blue), and (6,5) nanotube wrapped by one guanine (green).

**Figure 6 molecules-28-02719-f006:**
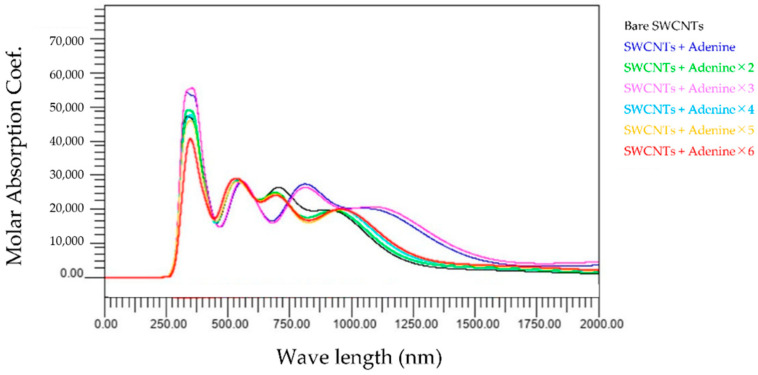
Wavelength dependence of light of the molar absorbance coefficient of adenine-wrapped SWCNTs.

**Figure 7 molecules-28-02719-f007:**
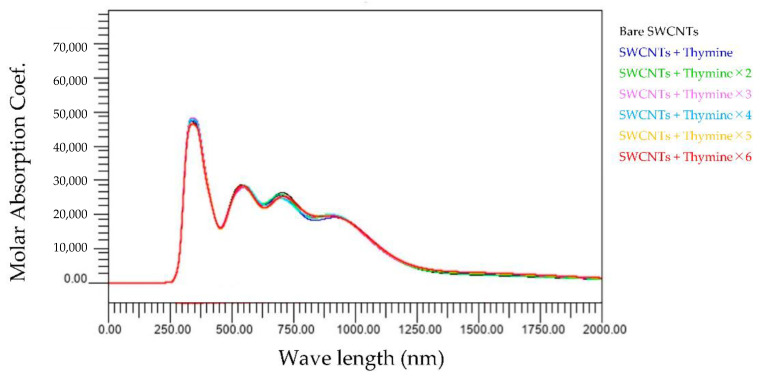
Wavelength dependence of light of the molar absorbance coefficient of thymine-wrapped SWCNTs.

**Figure 8 molecules-28-02719-f008:**
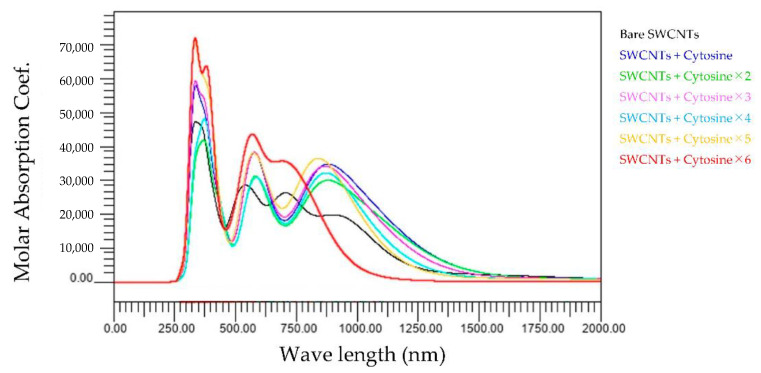
Wavelength dependence of light of the molar absorbance coefficient of cytosine-wrapped SWCNTs.

**Figure 9 molecules-28-02719-f009:**
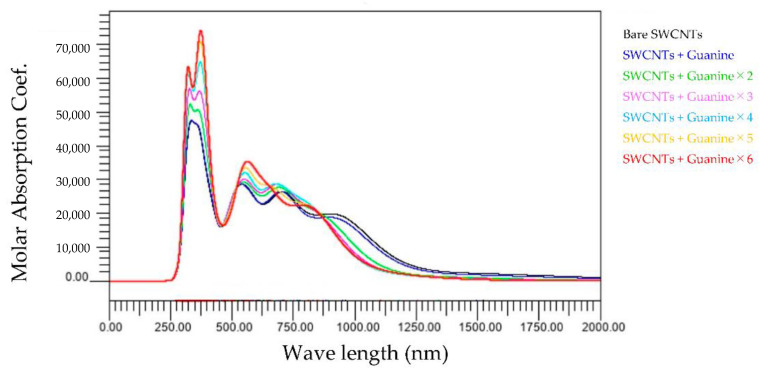
Wavelength dependence of light of the molar absorbance coefficient of guanine-wrapped SWCNTs.

**Figure 10 molecules-28-02719-f010:**
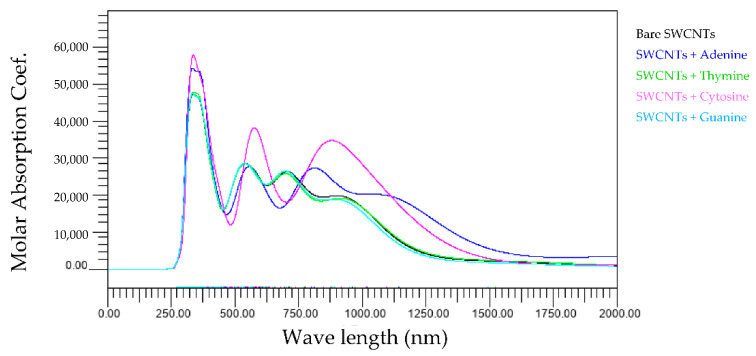
Wavelength dependence of light of the molar absorbance coefficient of bare- (black), one adenine- (blue), one thymine- (green), one cytosine- (purple), or one guanine (light-blue)-wrapped SWCNTs.

**Figure 11 molecules-28-02719-f011:**
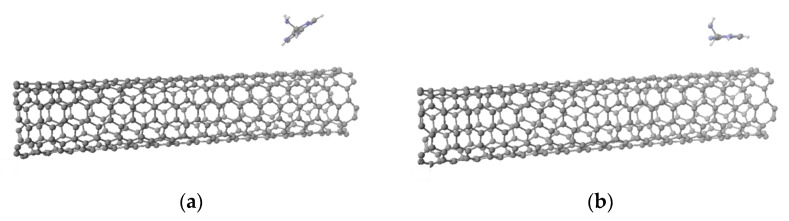
(**a**) Not-optimized molecular arrangement of (6,5) CNT and adenine. (**b**) Counterpart of optimized arrangement.

## Data Availability

Data are contained within the article.

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
