# Peer review of "Numerical Analysis for Light Absorption Spectra of the Base of DNA-Wrapped Single-Walled Carbon Nanotubes"

_molecules, 2023, doi:10.3390/molecules28062719_

Round 1
Reviewer 1 Report (Previous Reviewer 1)
The suggested changes have been addressed, so I consider that the paper can be accepted for publication in Molecules.
Author Response
Response to Reviewer 1 Comments
Point 1: The suggested changes have been addressed, so I consider that the paper can be accepted for publication in Molecules.
Response 1: We appreciate your positive comment.
Reviewer 2 Report (New Reviewer)
Dear,
The graphs of UV-Vis absorbance are poor quality, thats why is very difficult of the reader to understand the concept of each measurement. For this reason, I would like to see again the UV-Vis of the blank SWNTs dispersion, and after, the SWNTs dispersion with one of DNA. In this way, I think it could be easier for everyone to follow the research.
Also, what means 6_5_0/ 6_5_1 etc?
Author Response
Response to Reviewer 2 Comments
Point 1: The graphs of UV-Vis absorbance are poor quality, thats why is very difficult of the reader to understand the concept of each measurement. For this reason, I would like to see again the UV-Vis of the blank SWNTs dispersion, and after, the SWNTs dispersion with one of DNA. In this way, I think it could be easier for everyone to follow the research.
Response 1: If we correctly understand reviewer’s comments, the graphs of UV-Vis absorbance can be easily followed by the readers in the present overlapped style. As we have explained at page 4, the concept of each measurement is to compare the light absorption spectra of each structure. If our response to your comments does not satisfy you, further comments are appreciated. 
Point 2: Also, what means 6_5_0/ 6_5_1 etc?
Response 2: Thank you for your comment. 6_5_0/ 6_5_1 mean Bare SWCNTs/SWCNTs + Base. In this context, figure legends are amended as, for examples, “6_5_0” to “Bare SWCNTs”, “6_5_1” to “SWCNTs + Adenine” and so on.
We have used a paid editing service to improve our manuscript. Attached file is a confirmation certificate of English editing.

Round 2
Reviewer 2 Report (New Reviewer)
The overall article seems to be improved and thank for that, however, some parts are still under consideration.
The appearance of overall context of the images needs to be looked again
This manuscript is a resubmission of an earlier submission. The following is a list of the peer review reports and author responses from that submission.
Round 1
Reviewer 1 Report
The paper is very interesting and will undoubtedly provide very useful information to researchers working in this field. However, there are some observations that should be considered:
1.-The work is basically theoretical, but it should be related to experimental results. I think the authors should put more emphasis on comparing their findings with experimental results.
2.-Figures 1 to 5 should be improved. Perhaps one possibility would be to magnify the interaction zone between the bases and the nanotube.
3.-Figures 6 to 14 should be explained a little better. On the other hand, perhaps not all of them should be included in the paper. Some could go to supplemental material.
4.-The "Methods" section has no reference. Please include all the necessary references to conveniently cite the calculation method/procedure.
5.-In "Discussion" the authors should highlight more the relevance of the research and the possible applications for future work, especially from the experimental point of view.
Reviewer 2 Report
Taira and co-workers have used the SCIGRESS software package to model the adsorption of DNA bases on (6,5) SWCNT and the resulting changes in calculated OAS spectra. They employed the Lennard Jones potential for structure optimization and seem to have complete trust in a undisclosed black box method to calculate OAS spectra. Differences to experimental results are simply ascribed to differences between aqueous and vacuum environments. It is certainly reasonable, that a difference in dielectric background will affect excitonic optical transitions but not dipole active vibronic transitions. Albeit this discussion is absent from the manuscript. There is also no mention of the employed level of theory/approximation that was ,or rather had to be, picked in SCIGRESS. Why does thymine wrapping have a much weaker effect on the spectra than for instance cytosine wrapping? Can this effect be expected from the employed level of theory and the differences between the molecular structures?
In any way, adding the missing discussion should constitute a new submission.